



# Inferring the timing of abandonment of aggraded alluvial surfaces dated with cosmogenic nuclides

Mitch K. D'Arcy[1,2], Taylor F. Schildgen[2,1], Jens M. Turowski[2], Pedro DiNezio[3]

[1]Institute of Geosciences, University of Potsdam, Karl-Liebknecht-Strasse 24/25, 14476 Potsdam, Germany
[2]GFZ German Research Centre for Geosciences, Telegrafenberg, 14473 Potsdam, Germany
[3]Institute for Geophysics, Jackson School of Geosciences, University of Texas at Austin, J.J. Pickle Research Campus, Building 196 10100 Burnet Road (R2200), Austin, TX 78758, USA

*Correspondence to*: Mitch K. D'Arcy (mdarcy@uni-potsdam.de)

**Abstract.** Information about past climate, tectonics, and landscape evolution is often obtained by dating geomorphic
surfaces comprising deposited or aggraded material, e.g., fluvial fill terraces, alluvial fans, volcanic flows, or glacial till. Although surface ages can provide valuable information about these landforms, they can only constrain the period of active deposition of surface material, which may span a significant period of time in the case of alluvial landforms. In contrast, surface abandonment often occurs abruptly and coincides with important events like drainage reorganisation, climate change, or landscape uplift. However, abandonment cannot be directly dated because it represents a cessation in the deposition of
dateable material. In this study, we present a new approach to inferring when a surface was likely abandoned using exposure ages derived from in situ-produced cosmogenic nuclides. We use artificial data to measure the discrepancy between the youngest age randomly sampled from a surface and the true timing of surface abandonment. Our analyses simulate surface dating scenarios with variable durations of surface formation and variable numbers of sample exposure ages. From our artificial data, we derive a set of probabilistic equations and a Matlab tool that can be applied to a set of real sampled surface
ages to estimate the probable period of time within which abandonment is likely to have occurred. Our new approach to constraining surface abandonment has applications for geomorphological studies that relate surface ages to tectonic deformation, past climate, or the rates of surface processes.

## 1 Introduction

Geomorphological studies that link the formation of landforms to past changes in climate or tectonic deformation depend on
the accurate dating of surfaces comprising aggraded or deposited material. Surfaces commonly targeted for dating include alluvial fans, fluvial fill terraces, glacial till, pediments, and volcanic flows, among others. For example, fluvial fill terraces and alluvial-fan surfaces are widely dated in order to (i) decipher how erosion and sedimentation have responded to past hydroclimate changes (Owen et al., 2014; Schildgen et al., 2016; Tofelde et al., 2017); (ii) derive time-integrated slip rates for active faults (e.g., Frankel et al., 2007, 2011; Gosse, 2011; Hughes et al., 2018); and (iii) quantify the rates of surface



processes such as weathering, landform erosion, or channel avulsion and incision (Schildgen et al., 2012; Regmi et al., 2014; Bufe et al., 2017; D'Arcy et al., 2018).

A common assumption is that a geomorphic surface can be represented by a single formation age. Surfaces are usually point-sampled in multiple locations, e.g., by cosmogenic nuclide exposure dating of surface boulders. Typically, a limited number
(often fewer than 10) large, stable surface boulders are sampled for exposure dating, which exhibit no evidence of weathering, rotation, or disturbance. From the set of ages obtained, an average surface age is calculated with an uncertainty that reflects the spread of sampled ages. However, many geomorphic surfaces are active for an extended period of time, during which material is continually deposited until the surface is abandoned (e.g., Savi et al., 2016; Denn et al., 2017; Foster et al., 2017). Alluvial-fan surfaces provide one example. Rather than being formed instantaneously, fan surfaces are
typically active for thousands or tens of thousands of years before being abandoned when the channel avulses or incises (e.g., Dühnforth et al., 2007). This prolonged period of activity results in a meaningful spread in ages collected from a single surface (e.g., , Owen et al., 2011). For any geomorphic surface with a non-negligible period of formation, a set of surface ages will capture a portion of the full timespan over which that surface was active. An average of those ages will sit somewhere within the true timespan of surface deposition, but will overlook information such as the maximum age, which
might approximate the onset of surface activity, or the minimum age, which might approximate the timing of surface abandonment.

In some cases, the timing of surface abandonment may be a more useful constraint than an average surface age. In contrast to surface deposition, abandonment occurs at a particular moment in time (e.g., coinciding with a switch to incision) and so can, in principle, be defined with greater precision. For surfaces with an extended period of formation, the timing of
abandonment is more likely to coincide with events of interest such as reorganisation of a drainage network (Bufe et al., 2017); changes in climate, sediment supply, or base level (Steffen et al., 2009; Tofelde et al., 2017; Mouslopoulou et al., 2017; Brooke et al., 2018); or tectonic deformation such as faulting, uplift, or subsidence (e.g., Frankel et al., 2007, 2011; Ganev et al., 2010). Abandonment ages would also benefit any study that uses surface exposure dating to measure the rate of a post-depositional process, such as in situ weathering (e.g., White et al., 1996, 2005; D'Arcy et al., 2015, 2018), the
topographic decay of landforms (e.g., Hanks et al., 1984; Andrews & Bucknam, 1987; Spelz et al., 2008), or channel avulsion and incision (e.g., Schildgen et al., 2012; Finnegan et al., 2014; Malatesta et al., 2017). Yet the abandonment of a surface represents a cessation in the deposition of dateable material, and therefore cannot be directly dated. Instead, the timing of abandonment must be inferred. Some studies make assumptions about when geomorphic surfaces were abandoned based on independent information such as palaeoclimate records (e.g., Cesta and Ward, 2016); others assume that the
youngest sampled surface ages fall close to the timing of surface abandonment (e.g., Sarıkaya et al., 2015; Foster et al., 2017; Ratnayaka et al. 2018; Clow et al., 2019). These approaches risk circular or inaccurate interpretations, highlighting the need for a robust method to quantitatively infer the timing of surface abandonment from a set of sampled surface ages.

Here, we introduce a new probabilistic approach to constraining when a depositional surface was abandoned, based on what is known about its activity. We use artificial data to randomly point-sample the ages of virtual surfaces, in scenarios that are




representative of studies dating natural geomorphic landforms such as alluvial fans. We quantify how close the youngest obtained age is likely to fall to the true timing of abandonment, depending on the overall period of surface activity and the number of samples collected. From these artificial data, we derive a set of probabilistic equations and a Matlab tool that can be applied to real geomorphic surfaces to estimate when they were abandoned.

## 5   2 Justification

Here, we present a hypothetical example of a dated alluvial-fan surface to illustrate why the timing of abandonment may, in some cases, be more useful than an average of sampled surface ages.

Consider an alluvial-fan surface that was active for a 30 kyr timespan, starting at 80 ka and ending at 50 ka when the surface was abandoned due to fan incision (Fig. 1). In this example, the fan surface was deposited throughout a period of climatic

stability and abandoned when the climate changed, and we make the assumption that there is an equal likelihood of obtaining any age within the entire period of deposition. A distribution of surface ages can be obtained by point-sampling the fan surface; an approach analogous to studies using cosmogenic nuclides to measure the exposure ages of boulders. We present two possible outcomes in Fig. 1, where 6 surface ages are obtained. In scenario 1, the ages are distributed relatively evenly through time, producing a mean age of 65.8 ka that closely approximates the true average surface age of 65 ka, and a

standard deviation of 10.5 kyr. In scenario 2, the ages obtained are unevenly distributed through time, producing a slightly older mean surface age (71.4 ka) and a smaller standard deviation (5.2 kyr). These scenarios are plotted against time in Fig. 1b as data points and kernel density plots, and they resemble equivalent natural datasets (e.g., Owen et al., 2014).

Sample set 2 is more tightly clustered than sample set 1 despite being less representative of the average surface age, illustrating that greater clustering of ages is not an indicator of accuracy. Furthermore, neither average age corresponds to

any meaningful event. The fan surface was equally active for the entire period between 80 and 50 ka, the average ages sit within a period of climatic and depositional stability, and the peaks in the kernel density plots are artefacts created by randomly sampling a linear series.

In contrast, the abandonment of the fan surface does occur at a precise moment in time when deposition ends at 50 ka. In this example, abandonment coincides with an abrupt change in climate that triggered an incision event (cf., Simpson and

Castelltort, 2012), so is arguably a more informative target for dating than an average age that imprecisely approximates the mid-point in the duration of surface deposition. However, the abandonment of the surface represents a cessation in the deposition of dateable material, so its timing instead must be inferred from what is known about the surface activity. Given that the sampled ages constrain the timespan over which the surface was formed, and abandonment occurred sometime after the youngest age, it could be assumed that the youngest sampled age best approximates abandonment. In scenario 1, the

youngest age falls within ~1 kyr of surface abandonment, which would enable a correct interpretation of correlation between fan incision and the climate change event. In scenario 2, however, there is a ~14 kyr discrepancy between the youngest sampled age and the timing of surface abandonment, which would probably fail to demonstrate the correlation between





climate change and fan incision. Therefore, the question becomes: how close is the youngest age obtained from a surface to the actual timing of surface abandonment?

This question cannot currently be answered, yet the ability to reliably estimate when a surface was abandoned has important implications for many geomorphological studies (see section 1). In this study, we use artificial data to constrain the likely

time difference between the youngest age obtained from a geomorphic surface and the true timing of surface abandonment. There are several advantages to taking an artificial data approach. First, we can repeat the random sampling of surface ages (e.g., as depicted in Fig. 1) a large number of times to probabilistically determine where the youngest sampled age tends to fall. Second, we can prescribe the surface parameters, meaning the exact timing of abandonment and the full period of surface activity are known. Third, we can select surface properties that are representative of real geomorphic surfaces and

numbers of samples commonly obtained in geomorphic studies. Fourth, we can perform a thorough quantification of the uncertainties in our analyses.

## 3 Methods

### 3.1 Artificial data approach

We used artificial data to constrain the temporal discrepancy between the youngest age sampled on a surface and the actual

timing of surface abandonment. Our experiments are designed to be representative of natural alluvial-fan surfaces, but the results are more widely applicable to any abandoned depositional surface that has been subsequently dated.

In the absence of additional information (e.g., the existence of an additional surface with an intermediate age), the abandonment of a surface could have occurred at any time between the youngest sampled age, $a_{min}$, and the present, or within a particular time window after $a_{min}$. In the example case (Fig. 1), the data in sample set 1 would require a time

window of 1.1 kyr, (and 14.4 kyr for sample set 2), placed immediately after the youngest sampled ages, to overlap with the correct timing of surface abandonment, $t_{aban}$. We know the temporal discrepancy between $a_{min}$ and $t_{aban}$ in these examples because we impose $t_{aban}$; for real-world cases, this information is unknown. The likely proximity of $a_{min}$ to $t_{aban}$ will depend on the number of surface ages obtained, $n$. The greater the number of samples, the closer the youngest sampled age is likely to come to the abandonment age (Fig. 2a). The proximity of $a_{min}$ to $t_{aban}$ also depends on the total duration of

surface activity, which we denote as $T$. If $n$ ages are randomly-sampled from a longer time span, then $a_{min}$ is likely to fall farther from $t_{aban}$ (Fig. 2b).

Our artificial data experiments simulate surfaces with a length of the period of activity, $T$ between 10 and 50 kyr, sampled with numbers of surface ages, $n$ between 2 and 10. These values are representative of natural alluvial-fan surfaces and typical dating studies involving a small number of ages. For each combination of $T$ and $n$, we randomly sampled a set of surface

ages 10,000 times, allowing us to reliably constrain the probability that $a_{min}$ falls within a certain temporal distance of $t_{aban}$ in each scenario.





### 3.2 Implementation

We first implemented our experiments using discrete sampling within a spreadsheet. For each surface, we created a list of selectable surface ages spanning the total period of surface activity, $T$, and placed at equal intervals of 0.1 kyr. For the example case (Fig. 1), this would mean a list of selectable ages of 80.0 ka, 79.9 ka, 79.8 ka, etc., to a minimum value of 50.0
ka. We chose periods of surface activity, $T$, equal to 10, 20, 30, 40, and 50 kyr. From each list, we randomly selected $n$ unique values, and repeated this exercise 10,000 times for each integer value of $n$ between 2 and 10. For example, if $n = 6$ and $T = 20$ kyr, then we extracted 10,000 different datasets, each comprising 6 randomly sampled surface ages, from the 20 kyr-long list of selectable ages available at 0.1 kyr intervals. This process is analogous to simple random sampling of 6 cosmogenic nuclide exposure ages, e.g., from surface boulders, on an alluvial fan surface that formed over a 20 kyr period
and deposited a 'selectable' boulder every 100 years. We extracted 10,000 sets of surface ages for each of the 45 different combinations of $T$ (5 unique values) and $n$ (9 unique values). For each dataset, we calculated the mean value of the sampled ages, $\bar{a}$, and the time difference $a_{min} - t_{aban}$.

We define this time difference $a_{min} - t_{aban}$ as $\tau$, and this parameter is the primary focus of our analyses. From our artificial data, we extract cumulative frequency distributions of $\tau$ in each $T$-$n$ scenario.
To test whether 10,000 iterations are sufficient to produce reliable statistics and whether the discretization of ages has an important effect, we repeated all experiments using a non-discrete approach in a Matlab script. We defined $T$ as a time range, from within which any point in time could be randomly sampled, i.e., an excess number of 'selectable' surface ages were available rather than a list of discrete values. Performing 100,000 iterations with the Matlab script produced identical results to the discrete spreadsheet-based approach with 10,000 iterations. All data analyses are provided by D'Arcy et al. (2019) in
an online data repository. Finally, we explore the assumptions and limitations of our analyses in section 5.3.

### 3.3 Experimental assumptions

In designing our artificial data experiments, we make several assumptions. First, surface ages are randomly selected from the total period of surface activity. Therefore, when constructing our experiments, we assume that when ages are collected from real geomorphic surfaces, they are randomly point-sampling the full timespan of surface formation, and that this timespan
represents a uniform probability distribution of selectable ages. This may not always be the case, for example, if boulders on an alluvial-fan surface are spatially clustered by age and all samples are taken from one part of the surface. Second, the entire period of surface activity is assumed to be available for sampling, i.e., no subset of the surface history is missing as a result of processes like burial or erosion. Third, all selectable ages within the period of surface activity have an equal likelihood of being sampled; this implies that the surface formed with a constant deposition rate and there are no pulses of activity that
increase the probability of sampling a particular age. Finally, we do not explicitly factor in processes like nuclide inheritance, erosion, or incomplete exposure, which can affect exposure ages derived from cosmogenic nuclides. We consider the implications of all these assumptions for real datasets in section 5.3.



## 4 Results

### 4.1 Random sampling of surface ages

To illustrate the results of our experiments, we first present one example scenario in Fig. 3, in which the surface is formed

between 80 ka and 50 ka (i.e., $T = 30$ kyr) and is randomly sampled with $n = 2, 4, 6$, or 8 ages (with 10,000 repeat experiments for each value of $n$). Figure 3a shows how a frequency distribution of the mean value of all sampled ages, $\bar{a}$, changes with $n$. The distribution is centred on the true average surface age of 65 ka and narrows as a greater number of ages are sampled. If only 2 ages are sampled then $\bar{a}$ can occupy almost any age within the full period of surface activity, and as $n$ increases, $\bar{a}$ tends to fall closer to 65 ka. The distribution of $\bar{a}$ approaches a normal distribution as $n$ increases. This

observation is compatible with the central limit theorem and the law of large numbers, and $\bar{a}$ converges on the true average surface age as the number of samples increases, despite the dataset randomly sampling a linear series.

A frequency distribution can also be plotted for the youngest age, $a_{min}$, randomly sampled from the surface (Fig. 3b). If only 2 ages are collected, then the youngest can fall almost anywhere between 50 and 80 ka, although the distribution is asymmetric and younger values of $a_{min}$ occur slightly more frequently than older values. As $n$ increases, the distribution of

$a_{min}$ shifts towards 50 ka such that when $n = 8$, $a_{min}$ falls within 5-10 kyr of $t_{aban}$ in the majority of sampling experiments. As $t_{aban}$ is known in our experiments (50 ka), $\tau$ can be calculated for each set of ages sampled. Cumulative frequency distributions of $\tau$ reveal how close the youngest sampled age comes to the known timing of surface abandonment (Fig. 3c). For example, if only 2 ages are obtained, then in 60% of experiments $\tau \leq 12$ kyr, i.e., the youngest age falls somewhere within 12 kyr of abandonment. If 6 ages are obtained, then in 90% of experiments $\tau \leq 10$ kyr. Any percentile of $\tau$ can be

measured from Fig. 3c, allowing $\tau$ to be plotted against $n$ (Fig. 3d). As a greater number of ages are sampled, the value of $\tau$ associated with a given percentile decreases, i.e., the youngest sampled age comes closer to the timing of surface abandonment as the number of samples increases. However, the decrease in $\tau$ is non-linear and diminishes with increasing $n$. For example, as $n$ increases from 2 to 4 ages, the 95[th] percentile of $\tau$ falls from ~23 kyr to ~16 kyr, but collecting another 2 ages ($n = 6$) only reduces $\tau$ to ~12 kyr. The 95[th] percentile of $\tau$ falls below 10 kyr when $n$ exceeds 7 ages. In other words,

if 7 ages are randomly-sampled from a surface, abandonment will have occurred within 10 kyr after the youngest age in 95% of cases.

We equate the percentiles of $\tau$ in Fig. 3c with the probability, $P$, of abandonment occurring within a time window defined by $\tau$. Thus, if $P = 0.9$, the window of time $\tau$ (placed immediately after $a_{min}$) is large enough that in 90% of our experiments, the true timing of surface abandonment would fall within it. This is equal to the 90[th] percentile of $\tau$, which would be 7.5 kyr

for the scenario $T = 30$ kyr and $n = 8$, for example (Fig. 3d). Note that in this scenario, $\tau$ does not imply that the surface was abandoned exactly 7.5 kyr after $a_{min}$, but rather that there is a 90% likelihood that abandonment occurred anywhere within a





7.5 kyr window after $a_{min}$. The probable window of abandonment, $\tau$, increases with $P$ because a larger window of time is required to capture the true timing of abandonment in a greater proportion of cases. At the same time, $\tau$ is inversely and non-linearly related to the sample size of ages obtained, $n$ (Fig. 4). The dependencies between $\tau$ and $n$, $T$, and $P$ are illustrated in Fig. 4 for all tested scenarios that are representative of natural alluvial fan surfaces ($n$ = 2 to 10; $T$ = 10 to 50 kyr), with

probabilities between 0.50 and 0.95.

For example, if 6 ages are obtained from a surface that formed over a 30 kyr duration, then $\tau$ = 12 kyr for $P$ = 0.95 (Fig. 4a). If $P$ decreases to 0.5 (Fig. 4f) then $\tau$ decreases to 3 kyr for this particular scenario ($n$ = 6 and $T$ = 30 kyr).

The results of our artificial data experiments (Fig. 4) can be described by one equation that allows $\tau$ to be calculated for any scenario:

$$\tau = \tau_0 + PTe^{kn} \tag{1}$$

Here, the parameter $k$ is a decay constant that increases exponentially with $P$ (Fig. 5a):

$$k = a + be^{cP} \tag{2}$$

Constants $a$, $b$, and $c$ can be derived empirically using our artificial data. Note that we calibrate all our equations with time in kyr.

$a$ = -0.425 ± 0.029 $\qquad\qquad$ $b$ = 0.011 ± 0.011 $\qquad\qquad$ $c$ = 2.830 ± 0.885

The parameter $\tau_0$ increases linearly with $T$, but with a slope that increases exponentially with $P$ (Fig. 5b), and can therefore be described by a pair of relationships:

$$\tau_0 = mT \tag{3}$$

$$m = m_0 + ge^{hP} \tag{4}$$

Parameters $m_0$, $g$, and $h$ are constants with values again determined empirically from our artificial data experiments:

$m_0$ = 0.019 ± 0.008 $\qquad\qquad$ $g$ = 0.005 ± 0.002 $\qquad\qquad$ $h$ = 3.784 ± 0.406

Given that $\tau_0$ signifies the value of $\tau$ as $n$ trends towards infinity, it represents the most precise possible constraint on the abandonment period, $\tau$. For the scenarios shown in Fig. 5b, which represent reasonable values of $T$ for natural alluvial fans and desirable values of $P$, $\tau_0$ varies from a few centuries to ~10 kyr. These $\tau_0$ values illustrate the limits to precision when

inferring the timing of surface abandonment in this probabilistic way.

## 4.2 Total period of surface formation

Equations 1-4 can be solved for $\tau$ with knowledge of only the number of ages sampled, $n$, and the total period of surface formation, $T$, as well as a chosen probability, $P$. We are able to parameterise equations 1 through 4 using artificial data because we know the value of $T$ in our experiments. However, when sampling real depositional surfaces, $T$ is unknown and

instead only the span of sampled ages, $a_{max} - a_{min}$, can be measured. This span might approximate $T$, but some fraction of time will remain unsampled. To resolve this conundrum, our artificial data experiments also determine the fraction of $T$ that is captured by $a_{max} - a_{min}$ in scenarios of varying $n$ (Fig. 6).





The data indicate that, for example, 6 randomly-distributed ages will span ~70% of the total timespan of surface activity, $T$, in the average case. In the 1% of 'worst' (most clustered ages) cases, $a_{max} - a_{min}$ will only represent ~30% of $T$, and in the 1% of 'best' (least clustered ages) cases it will represent more than 95% of $T$. In half of all experiments for $n = 6$ (from P25 to P75), $a_{max} - a_{min}$ falls within 60-85% of $T$. There is a diminishing improvement with an increasing number of sampled

ages, such that by $n = 10$, the average span of ages has only increased to ~80% of $T$ and 50% of all experiments fall between 75-90% of $T$. An order of magnitude more ages (hundreds) would be needed for the mean $a_{max} - a_{min}$ to come within 95% of the full period of surface activity.

A regression can be fitted to the distributions in Fig. 6, taking the form:

$$\frac{a_{max} - a_{min}}{T} = q + re^{sn} \tag{5}$$

Parameters $q$, $r$, and $s$ are empirical coefficients derived graphically from our artificial data. For the mean case (the solid black line in Fig. 6), they take the values:

$q_{av} = 0.838 \pm 0.007$          $r_{av} = -1.035 \pm 0.030$          $s_{av} = -0.366 \pm 0.016$

Equation 5 is also fitted to ±1 standard deviation ($\sigma$) above and below the mean values in Fig. 6 (dashed black lines). For $1\sigma$ above the mean, parameters $q$, $r$, and $s$ take the values:

$q_{+1\sigma} = 0.928 \pm 0.005$          $r_{+1\sigma} = -0.983 \pm 0.055$          $s_{+1\sigma} = -0.512 \pm 0.027$

For $1\sigma$ below the mean, parameters $q$, $r$, and $s$ take the values:

$q_{-1\sigma} = 0.764 \pm 0.007$          $r_{-1\sigma} = -1.196 \pm 0.015$          $s_{-1\sigma} = -0.296 \pm 0.008$

Equation 5 can therefore be used to estimate the size of $T$ in the average case plus ±$1\sigma$ bounds, given the measured span of ages collected from a surface. Equations 1-5 can now be used to probabilistically calculate a window of time during which

any dated surface was likely abandoned.

**4.3 Application to real surface ages**

Given that Eq. 1 is probabilistic (i.e., $P$ is a variable), it can be used to infer a probability distribution of abandonment ages from a set of measured surface ages. We illustrate the steps involved in applying Eq. 1 to real data in Fig. 7.

First, $\tau$ is calculated using equations 1-5 with discrete values of $P$ (Fig. 7a), resulting in discrete windows of time in which

abandonment is likely to have occurred with different probabilities. These values of $\tau$ can be converted into a probability distribution (Fig. 7b) by calculating the density of $P$ within fixed increments of $\tau$. If the sampled surface ages are known with exact precision, then the resulting distribution of $\tau$ values provides a probability distribution of times that would directly postdate the youngest age and yield a probability distribution of surface abandonment ages. However, real surface ages have associated uncertainties that must also be incorporated into the estimated abandonment ages (Fig. 7c). First, we use ±$3\sigma$

uncertainty on $a_{min}$ to determine a probability distribution of potential $a_{min}$ values; in Fig. 7c we assume a normal distribution, as is typical for exposure ages of individual boulders, but alternative distributions could be used if appropriate. This distribution of $a_{min}$ values is then discretised, and separate probability distributions of $\tau$ are calculated for each



potential value of $a_{min}$, i.e., repeating Fig. 7a-b. The resulting, temporally shifted probability distributions of $\tau$ are weighted according to the probability distribution of $a_{min}$ and summed, resulting in an overall probability distribution of likely abandonment ages that accounts for uncertainty on the youngest age (Fig. 7c).

If the $1\sigma$ uncertainty on $a_{min}$ is small compared to $\tau$ calculated using Eq. 1, then incorporating age uncertainty will have little impact on the resulting probability distribution of abandonment ages. If the $1\sigma$ uncertainty on $a_{min}$ is large, it will have a greater influence on the final probability distribution of abandonment ages. In the supporting information, we provide a Matlab script that can be used to input a set of measured surface ages and output a probability distribution of abandonment timings following the steps outlined in Fig. 7.

## 5 Discussion

### 5.1 Implications for surface dating

Our artificial data provide new information about what measured ages represent when collected from aggraded surfaces that formed over non-negligible timespans. Crucially, our findings indicate that averages of sampled surface ages are likely to be imprecise representations of the mid-point of surface formation, and may not correlate with any external forcing event (Fig. 1). In contrast, surface abandonment typically occurs at a discrete moment in time and is more likely to coincide with external forcing events such as changes in climate or tectonics. By using artificial data, we have derived a set of probabilistic equations for inferring when a surface was likely to have been abandoned, based on a distribution of randomly-sampled surface ages. These equations can complement and enhance interpretations based on any dataset comprising surface ages.

While a distribution of ages is required for dating surfaces that have formed over extended periods of time, our analyses reveal diminishing returns from sampling an increasing number of ages; these diminishing returns apply to constraints on abandonment (Figs 3d and 4) and the total duration of surface activity (Fig. 6). An appropriate number of surface ages will depend on the desired precision, but our results indicate that there is little to be gained by obtaining more than 6 to 7 ages per surface (Figs. 3, 4, and 6), assuming no outliers, for the purposes of most geomorphological studies. Indeed, to obtain significantly more information about a surface, an order of magnitude more ages would be required. As explained in section 4.1, $\tau_0$ represents the maximum precision with which the abandonment age can, in principle, be inferred. For many natural surfaces, $\tau_0$ can range from a few centuries to ~10 kyr (Fig. 5d), depending on the period of surface activity and the desired probability. Our methodology thus provides a new way of quantifying the limits to the precision with which a distribution of surface ages can be interpreted. These limits are in addition to the age uncertainty associated with cosmogenic nuclide exposure dating, and both are an important consideration when inverting landforms and sedimentary deposits for palaeo-environmental information (Foreman and Straub, 2017).

When sampling in the field, it may sometimes be advantageous to target different parts of a surface in order to capture as much of its period of activity as possible. This strategy applies to surfaces upon which the locus of deposition has systematically migrated during deposition. For example, if channel migration on an alluvial-fan surface resulted in particular



fragments of its overall history being recorded in particular parts of the surface (e.g., Savi et al., 2016; Schürch et al., 2016; D'Arcy et al., 2017a,b), then greater spatial coverage would capture a greater range of ages. However, if each deposition event followed a random trajectory on the surface, resulting in all potentially 'selectable' ages being spatially mixed, then it would be unnecessary to distribute sampling locations across the surface.

## 5.2 Case study: Alluvial fans in the Laguna Salada Basin, Mexico

Here, we use a case study of alluvial-fan surfaces in the Laguna Salada Basin, Mexico, to demonstrate how our findings can be applied to real surfaces to gain new information about when they were abandoned.

The Laguna Salada Basin is a half-graben in northern Baja California, Mexico. This basin contains well-preserved alluvial fans eroded from the neighbouring Sierra El Mayor and Sierra Cucapa, with at least 8 generations of distinct fan surfaces formed by a sequence of aggradation and incision cycles. The ages of two of these fan surfaces—mapped as Q4 and Q7— were estimated by Spelz et al. (2008) using $^{10}$Be exposure ages of stable surface boulders with no evidence of erosion or disturbance (Fig. 8). We used the CREp calculator (Martin et al., 2017) to update the exposure age estimates of Spelz et al. (2008) using the time-corrected Lal/Stone scaling scheme (Lal, 1991; Stone, 2000), the ERA40 atmosphere model (Uppala et al., 2005), the atmospheric $^{10}$Be-based VDM geomagnetic database of Muscheler et al. (2005) and Valet et al. (2005), and the current global reference SLHL $^{10}$Be production rate of 4.13 ±0.20 at g$^{-1}$ yr$^{-1}$ in the ICE-D database (Martin et al., 2017). We assume a sample density of 2.7 g cm$^{-3}$ and no boulder erosion. The oldest age measured on the Q4 surface was excluded as an outlier by Spelz et al. (2008), and we maintain this interpretation. The remaining exposure ages span 14.4 ±1.1 ka to 32.1 ±2.9 ka for Q4 ($n = 5$), and 188.6 ±22.7 ka to 246.9 ±13.7 ka for Q7 ($n = 6$) (Fig. 8b, yellow bars). On both fan surfaces, the dispersion of ages greatly exceeds the age uncertainty, confirming that each surface was deposited over an extended period of time.

For both distributions of fan surface ages, we used equations 1 to 5 to calculate probable abandonment windows, $\tau$, for different values of $P$. For example, on the Q4 fan surface with an $a_{min}$ of 14.4 ±1.1 ka, $\tau = 3.3$ kyr when $P = 0.5$, suggesting a 50% probability that the surface was abandoned within 3.3 kyr after $a_{min}$, i.e., between 14.4 ka and 11.1 ka. The size of $\tau$ increases with $P$, as explained in section 4.1, such that $\tau = 12.0$ kyr for the Q4 surface when $P = 0.95$, i.e., the abandonment window ranges from 14.4 ka to 2.4 ka. If Eq. 1 is applied to the data without accounting for age uncertainty, the resulting probability distribution of $\tau$ is highly asymmetric (Fig. 8c, red dashed lines). Of course, the uncertainty on $a_{min}$ must be accounted for, so following the approach outlined in Fig. 7, we derived a continuous probability distribution of $\tau$ for the Q4 and Q7 fan surfaces that incorporates age uncertainty using the Matlab tool (Fig. 8c, solid black lines). These probability distributions of $\tau$ illustrate how the likelihood of surface abandonment is distributed over time for two representative natural datasets. On the Q4 surface, the measured age uncertainty is small compared to $\tau$, so the resulting $\tau$ distribution has an asymmetric shape that is primarily determined by the form of Eqs 1 to 5 and our artificial data calibration (Figs 3 and 4). The majority of the Q4 $\tau$ distribution occupies a short timespan that is smaller than the spread of sampled surface ages; this result supports our reasoning that the timing of surface abandonment can, in some cases, be constrained more precisely than a



representative age of surface formation (see section 2). The age uncertainty on $a_{min}$ is significantly larger on the older Q7 surface and therefore dominates the probability distribution of $\tau$, giving it a wider and more symmetrical shape despite the greater number of measured ages, $n$. This result underscores the importance of accounting for age uncertainty when using our equations to infer the likely timing of surface abandonment, which our supplementary Matlab tool incorporates.

### 5.2.1 Climatic implications

Our estimates of when the Laguna Salada fans were abandoned have important climatic implications. Spelz et al. (2008) speculated that the aggradation and incision of the fan surfaces was partly controlled by past climate changes, and there is growing evidence that alluvial systems can be highly sensitive hydroclimate recorders (D'Arcy et al., 2017a,b; Wickert and Schildgen, 2019). We explore this idea by comparing the surface age data with two palaeoclimate proxy records (Fig. 8): the GRIP ice core δ[18]O record from Greenland (Johnsen et al., 1997) and the LR04 global benthic δ[18]O stack (Lisiecki and Raymo, 2005). These records primarily reflect the growth and decay of continental ice sheets, which are generalised into Marine Isotope Stages (MIS).

The sampled Q7 ages clearly coincide with the broadly interglacial conditions of MIS 7, so we interpret that the surface was deposited throughout this stage. Our statistical analyses indicate that the Q7 surface was abandoned—in this case due to fan incision—during the subsequent MIS 6 and coinciding with a transition to a more glacial climate. Indeed, 71% of the area beneath the Q7 $\tau$ distribution falls within MIS 6 (191-130 ka), which we interpret as a 71% likelihood that the surface was abandoned and incised during this stage. For the Q4 fan surface, the sampled ages alone indicate that abandonment coincided with the end of the Last Glacial Maximum (MIS 2) and the global shift to interglacial conditions in the Holocene. Spelz et al. (2008) interpreted this observation (fan incision during a shift to interglacial climate) to contradict the Q7 data (fan incision during a shift to glacial climate). However, supplementing the measured ages with our probabilistic analyses reveals that Q4 abandonment is likely to have occurred during the Younger Dryas, a short-lived climate episode between 12.9-11.7 ka during which the northern-hemisphere climate returned to a cooler state (Carlson, 2013). In Fig. 8c, 36% of the $\tau$ distribution falls within the Younger Dryas and the peak of the $\tau$ distribution—i.e., the single most probable abandonment age—falls at 12.7 ka. This interpretation reconciles the Q7 and Q4 surfaces on the Laguna Salada fans, which would have both been incised as a result of climatic shifts towards more glacial conditions. This case study also demonstrates how our probabilistic approach, uniquely enabled by our use of artificial data, can be used to quantify the likelihood of individual abandonment scenarios and strengthen palaeoclimatic interpretations derived from alluvial deposits.

### 5.2.2. Tectonic and weathering implications

The results in Fig. 8 also have tectonic implications. The Laguna Salada fans are dissected by fault scarps related to the Laguna Salada fault and the Cañada David detachment; the largest Q7 scarp has an offset of 9.9 m (Spelz et al., 2008). Typically, studies divide the fault offset by the mean surface age (which for Q7 is 215.9 ka) to estimate a time-averaged slip rate, which would be 0.046 mm yr[-1] in this example. However, as a scarp can only accumulate displacement once the surface





has been abandoned, i.e., when it is no longer being resurfaced, the estimated age of abandonment may be a more appropriate timescale for determining a displacement rate. Accumulating a 9.9 m offset since 177 ka (the most likely abandonment age, Fig. 8c) would produce a time-averaged slip rate of 0.056 mm yr$^{-1}$; an increase of 22%. Following this logic, the probability distribution of $\tau$ could be translated into a probability distribution of time-averaged slip rates. For the Q4 fan surface, calculating a slip rate with a most likely abandonment age (e.g., 12.7 ka) instead of the mean surface age (23.3 ka) would result in an even larger increase in the calculated displacement rate of 83%. Underestimating fault slip rates by this magnitude could have major implications for tectonic and fault hazard analyses.

Spelz et al. (2008) also measured the diffusional decay of fault scarp geometry over time, and used the calculated mean fan surface ages to derive time-integrated scarp mass diffusivities between ~0.01-0.10 m$^2$ kyr$^{-1}$. Intriguingly, the authors interpreted these diffusivities to be anomalously slow. This conundrum could be partly resolved by, again, using the estimated surface abandonment ages to calculate scarp mass diffusivity, rather than average surface ages. This approach would result in faster diffusion rates, as Spelz et al. (2008) expected, while simultaneously recognising that a fault scarp can only form and erode once a fan surface has been abandoned.

The alluvial fans of the Laguna Salada Basin provide a representative example of natural, aggraded geomorphic surfaces, which are formed over a non-negligible period of activity and dated with a small set of exposure ages that randomly-sample the duration of surface activity. This case study demonstrates that Eqs 1 through 5, together with an incorporation of exposure age uncertainty provided by the Matlab tool, can provide valuable constraints on the timing of surface abandonment. These constraints complement the sampled surface ages and can enhance interpretations involving palaeoclimate, tectonics, and landform evolution.

## 5.3 Limitations to the probabilistic approach

Our artificial data, and therefore the parameterisation of Eqs 1 to 5, assume that a distribution of surface ages are randomly sampling the full duration of surface activity. In some cases, this assumption might be realistic, e.g., the Q7 surface on the Laguna Salada fans (Fig. 8) was sampled in different places and produced ages spanning all of MIS 7, suggesting the full duration of surface activity might be well-represented. If so, Eqs 1 to 5 could be symmetrically applied to the oldest sampled age to estimate the onset of deposition. In contrast, the Q4 surface was sampled entirely at the fan apex, where enhanced vertical aggradation makes it likely that the earliest deposits from this depositional episode have been buried. In practice, this sampling approach would improve estimates of when abandonment occurred. By clustering the surface ages towards the end of the depositional period, $T$ would effectively shorten and $\tau$ would be constrained more precisely. Because our approach derives $T$ empirically from the ages that are actually sampled on a surface (Eq. 5), the burial of early deposits does not matter for estimating abandonment. However, vertical burial would mean that $T$ would no longer represent the total duration of deposition, and it would therefore be inappropriate to use our equations to estimate the onset of deposition.

Like burial, subsequent erosion of part of a surface might hide a fragment of the period of deposition from sampling. The implications of erosion depend on how spatially-homogenous the surface is, i.e., whether erosion has randomly eliminated



'selectable' ages from throughout the duration of activity, or instead eradicated complete fragments of the timespan of activity. Again, erosion would only impede our method of inferring the abandonment age if the youngest part of the duration of activity were destroyed. Given that burial and erosion are site-specific, they cannot be universally incorporated into our equations and must be considered on an individual-case basis.

Equations 1 to 5 assume that all sampled surface ages are true ages. In reality, incorrect ages are sometimes encountered when dating surfaces. For example, cosmogenic nuclide exposure ages may be biased towards older ages as a result of nuclide inheritance, as is interpreted to be the case with the oldest age on the Laguna Salada Q4 fan surface (Fig. 8a). Including old outliers in our analyses would lead to an over-estimation of the size of both $T$ and $\tau$, and therefore unnecessarily imprecise estimates of the abandonment window, but would not change the position of $a_{min}$. A more serious

error would arise from incorrect young ages, e.g., resulting from erosion or shielding of boulders targeted for cosmogenic nuclide exposure dating. The inclusion of spurious young ages could expand the apparent period of surface activity $T$ past the true timing of abandonment, leading to estimates of $\tau$ that are both too large and, more importantly, too young. Therefore, equations 1-5 and the Matlab tool should be applied to 'clean' datasets that do not contain spurious ages, and particularly not spuriously young ages when attempting to calculate abandonment times.

Finally, our approach derives the true period of surface activity, $T$, from the measured age range $a_{max} - a_{min}$, based on the results of our artificial data experiments (see section 4.2 and Fig. 6). This step is necessary because the true duration of $T$ is ultimately unknowable for natural surfaces, so we parameterise Eq. 5 using the mean ratio of $(a_{max} - a_{min})/T$ among our artificial data experiments. Of course, any given set of real surface ages might happen to capture a greater or smaller fraction of $T$ than the mean case. For this reason, we also provide parameterisations of Eq. 5 for $\pm 1$ standard deviation above and

below the mean ratio of $(a_{max} - a_{min})/T$, thus allowing $\pm 1\sigma$ uncertainty on $T$ to be tested. In practice, the uncertainty associated with $T$ has little effect on the probability distributions of $\tau$ produced by Eq. 1, and so is likely to be insignificant for most geomorphological applications. To illustrate the sensitivity of $\tau$ to the uncertainty on $T$, we re-calculate the probability distributions of $\tau$ for the Q4 and Q7 Laguna Salada alluvial fan surfaces with the Matlab tool (Fig. 9) using the $\pm 1\sigma$ bounds on $T$ (Fig. 6).

The uncertainty on $T$ has a negligible effect on the probability distributions of $\tau$, for both the young and precisely-dated Q4 surface where the $\tau$ distribution is most sensitive to the form of Eqs 1 to 5, and the older, less-precisely dated Q7 surface where the $\tau$ distribution is most sensitive to the measured age uncertainty. This sensitivity analysis demonstrates how the conversion of $a_{max} - a_{min}$ to $T$ has little bearing on the estimated timings of surface abandonment. Nonetheless, our artificial data calibration allows the $\pm 1\sigma$ uncertainty on $T$ to be calculated, if desired.

**6 Conclusions**

Our study uses artificial data to simulate depositional geomorphic surfaces that form over a non-negligible timespan, and are subsequently dated with a set of randomly-sampled surface ages. We investigate scenarios that are representative of natural



alluvial fans, which are commonly targeted for surface dating, however our results may be more broadly applicable to other depositional landforms that form over protracted periods of time. Our findings suggest that, for a variety of different purposes, inferring the timing of surface abandonment may provide more informative and more precise interpretations than taking an average of measured surface ages. We use our artificial data to derive a set of probabilistic equations that can be

applied to a distribution of real sampled surface ages to estimate a period of time within which abandonment is likely to have occurred. These equations account for site-specific variables including the number of ages and the duration of activity for a particular surface, and they can be used to generate a probability distribution of likely abandonment ages. We furthermore provide a Matlab script that allows for the uncertainty associated with measured ages to be incorporated in the probability distribution of abandonment ages. The ability to constrain the timing of surface abandonment has useful applications for

geomorphological studies that relate surface ages to tectonic deformation (e.g., deriving fault slip rates), climate (e.g., reconstructing past hydroclimate changes), or the rates of surface processes (e.g., weathering and landform evolution), a subset of which we demonstrate using a case study of alluvial fan surfaces in the Laguna Salada Basin, Mexico. The statistical framework we introduce in this paper offers a new method of probabilistically estimating when a surface was abandoned, which can complement and enhance interpretations of any distribution of sampled ages obtained from surfaces

that experienced a non-negligible period of deposition.

## 7 Code availability

The Matlab script used to analyse the data is provided in the supplement.

## 8 Data availability

All data in this article are presented in the main paper and are freely available online via the Figshare repository at:
https://figshare.com/s/8001bf97556ae078d1e8.

## 9 Author contributions

MKD, JMT, and TFS conceived of the idea for this article. MKD performed the analyses and wrote the main text. TFS created the Probabilistic Sampling Matlab script and contributed to interpretations. JMT contributed to the statistical analyses and PDN contributed to climatic interpretations. All authors commented on and contributed to the final article.

## 10 Competing interests

They authors declare that they have no conflict of interest.

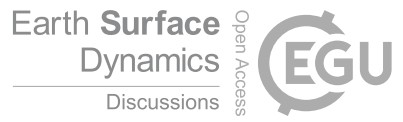

**7 Acknowledgements**

M. D'Arcy was supported by an Alexander von Humboldt postdoctoral fellowship, a Research Focus: Earth Sciences grant awarded by the University of Potsdam, and by the Emmy-Noether- Programm of the Deutsche Forschungsgemeinschaft grant number SCHI 1241/1-1 awarded to T. Schildgen.

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

10   **Figures**

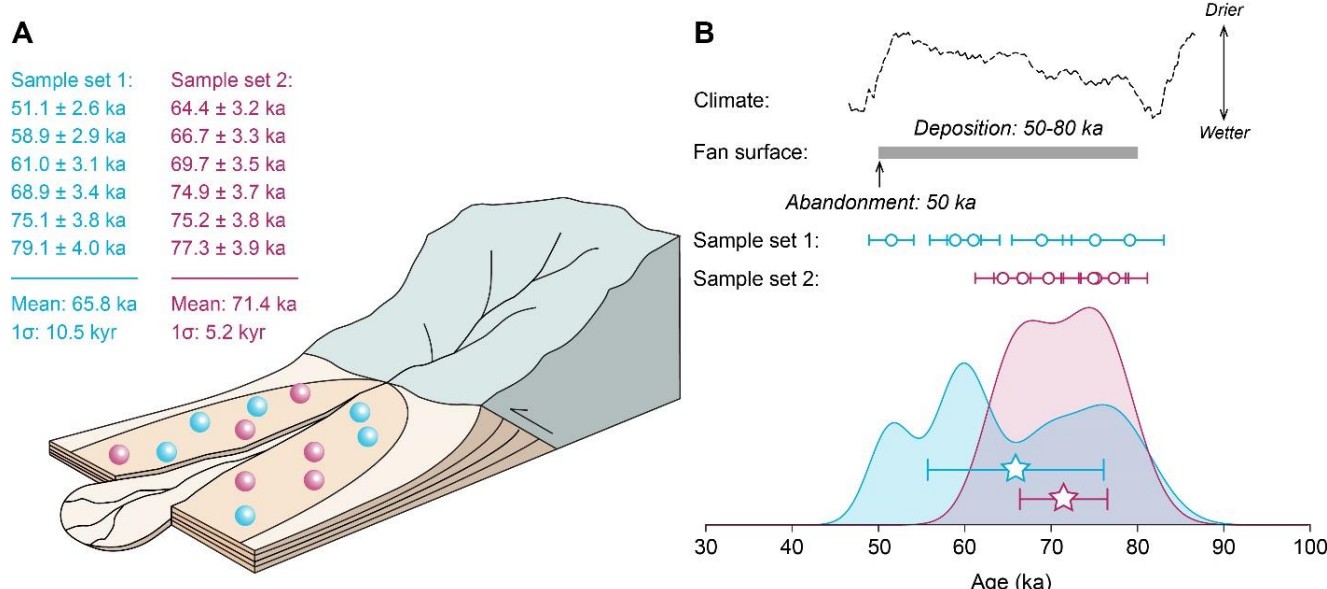

**Figure 1: (A) Conceptual alluvial fan surface that was formed over a 30 kyr period, from 80 ka to 50 ka, after which it was**
15   **abandoned, e.g., due to incision. Two different dating scenarios (sample sets 1 and 2) are shown in which 6 surface ages are randomly selected. (B) The true period of surface activity (grey bar), compared with the sampled ages presented as data points (circles), kernel density plots, and mean surface ages ±1 standard deviation (stars). A hypothetical climate scenario is depicted as a dotted line.**



Earth **Surface**
Dynamics
Discussions



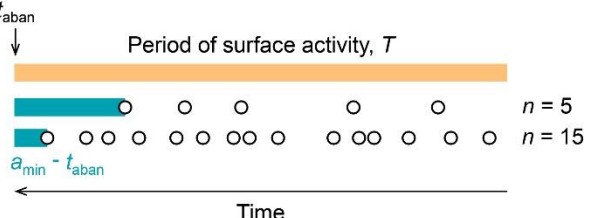

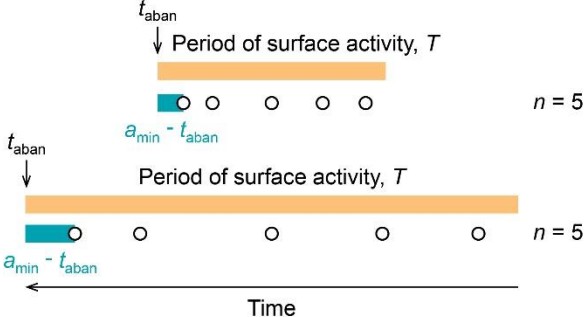

**Figure 2: Schematic surface with a period of activity, $T$ (orange bar), abandoned at $t_{aban}$, and randomly sampled with $n$ ages (circles). (A) If $n$ increases, the youngest sampled age, $a_{min}$, is likely to fall closer to $t_{aban}$. (B) If $T$ increases, the youngest sampled age, $a_{min}$, is likely to fall farther from $t_{aban}$, even if the same number of ages are sampled.**





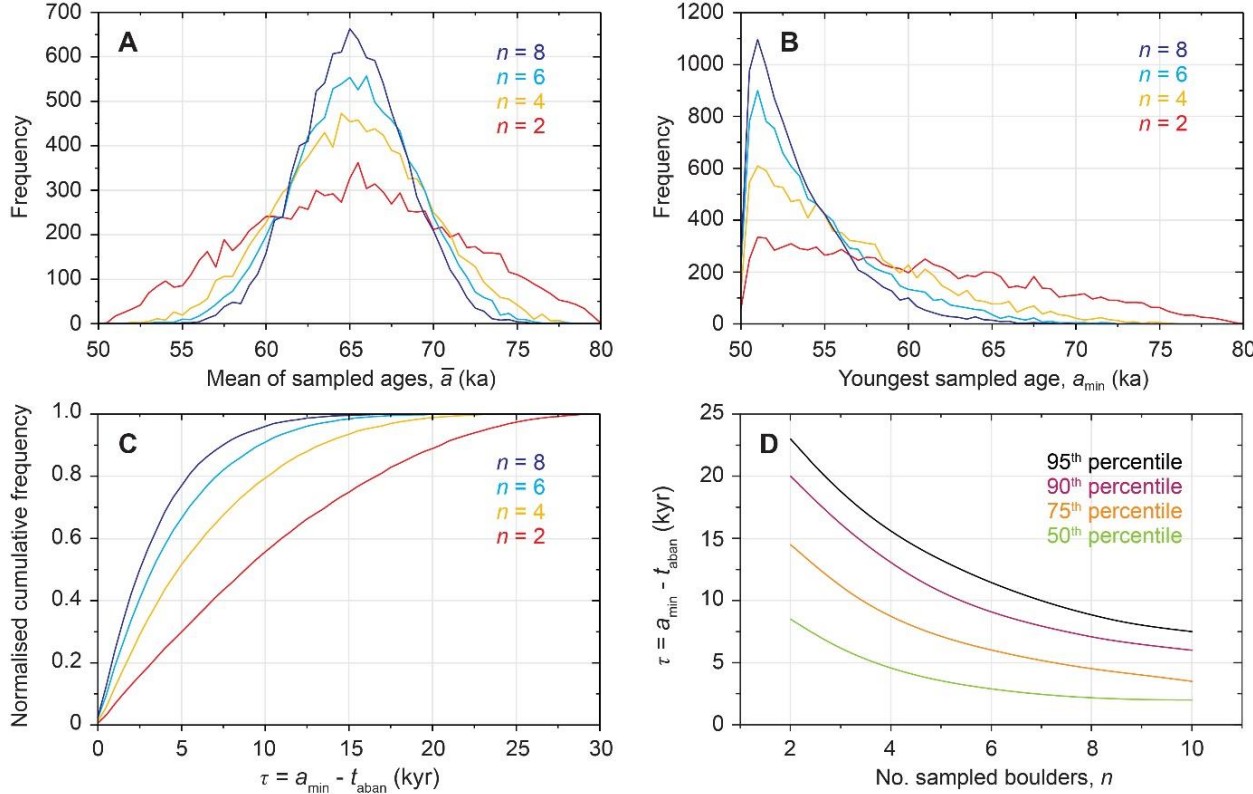

Figure 3: Example results of the artificial data experiments for a surface active from 80 to 50 ka ($T = 30$ kyr). A number of ages, $n$, were randomly sampled from the surface 10,000 times. (A) Frequency distributions of resulting mean sampled age, $\bar{a}$. (B) Frequency distributions of the youngest sampled age, $a_{min}$. (C) Cumulative frequency distributions of $\tau$ normalised to a sum of 1. (D) Selected percentiles of $\tau$ plotted against $n$.







**Figure 4: The probable abandonment window, $\tau$, as a function of the number of boulder ages, $n$. Data are shown for different probabilities, $P$ (panels), and durations of surface activity, $T$ (colours). Parameter $k$ is a decay constant that depends on $P$ (see text for details).**



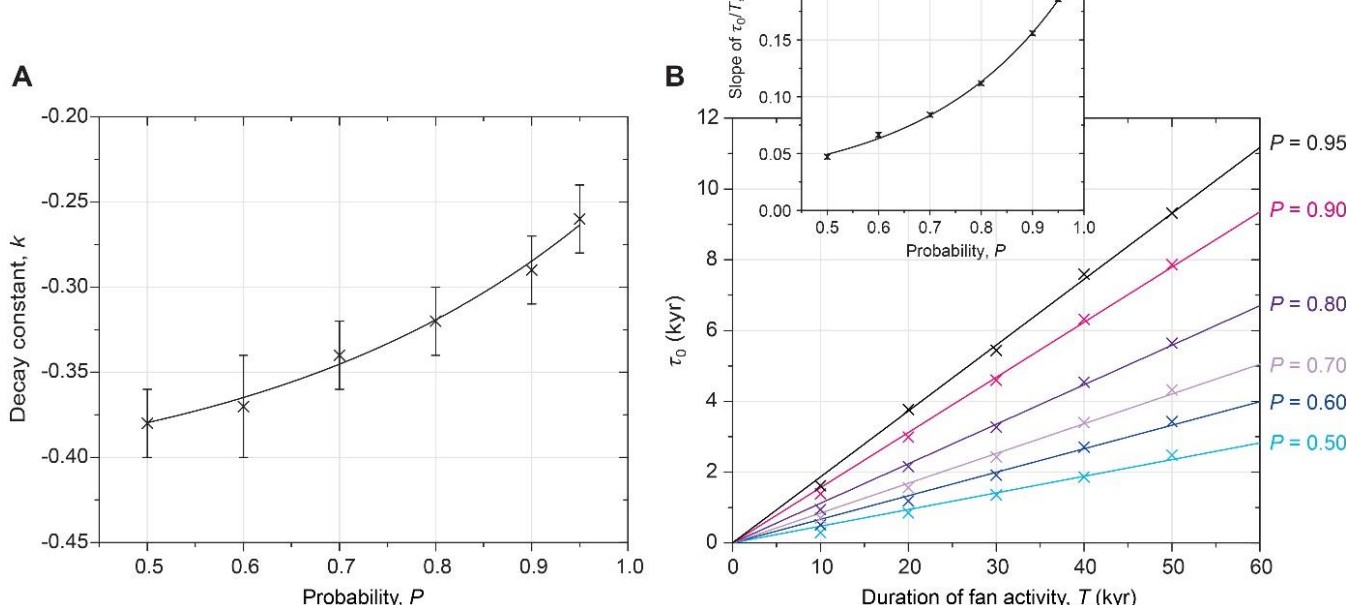

**Figure 5: (A)** Variation in the decay constant, $k$, as a function of the probability, $P$. Error bars show the standard error on $k$ when Eq. 1 is fitted to the data in Fig. 4. The regression corresponds to Eq. 2. **(B)** Variation in $\tau_0$ as a function of $T$ for different values of $P$ (indicated by colours). Linear regressions are fitted corresponding to Eq. 3. Inset: Variation in $m$ as a function of $P$. An exponential regression is fitted corresponding to Eq. 4.



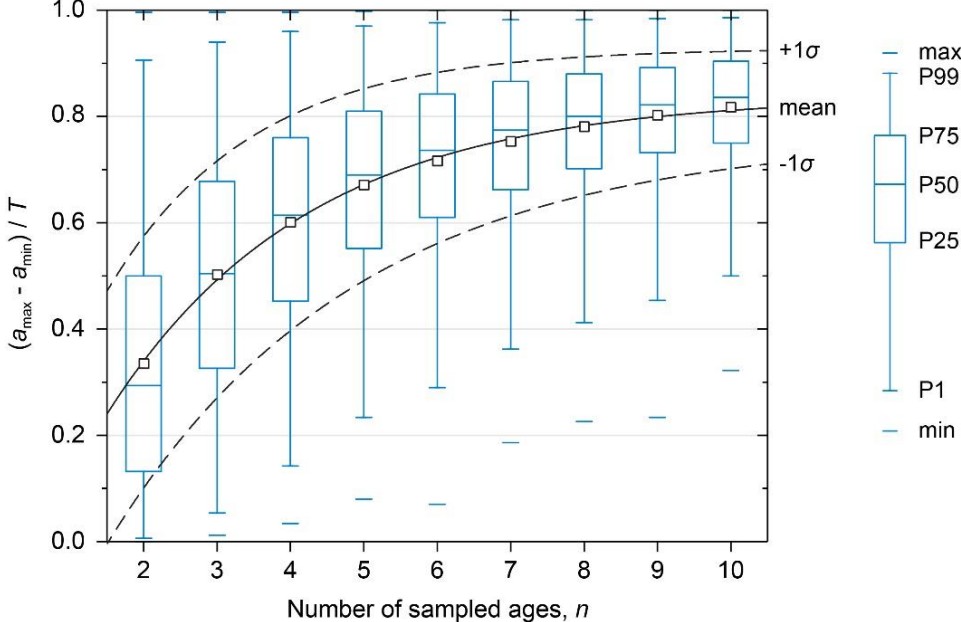

**Figure 6: Box plots showing the fraction of the total period of fan activity, *T*, captured by the span of sampled boulder ages in the artificial data experiments, $a_{max} - a_{min}$ plotted against the number of sampled ages, *n*. Each box represents 10,000 experiments. As a greater number of ages are sampled, the span of the set of ages is more likely to capture a greater fraction of *T*, although with diminishing returns for increasing *n*. Black lines show exponential regressions corresponding to Eq. 5 and fitted to the mean values (solid) and ±1 standard deviation (dashed).**

Earth **Surface**
Dynamics
Discussions
EGU
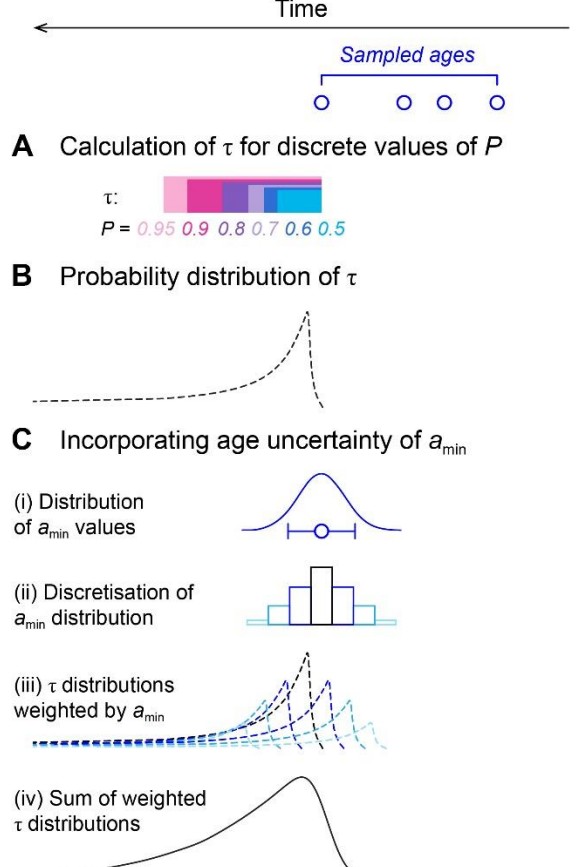

**D**  Equations

[1]  $\tau = \tau_0 + PTe^{kn}$

[2]  $k = a + be^{cP}$

[3]  $\tau_0 = mT$

[4]  $m = m_0 + ge^{hP}$

[5]  $T = \dfrac{a_{max} - a_{min}}{q + re^{sn}}$

**Figure 7: Schematic demonstrating how to infer the timing of surface abandonment from a set of sampled ages. (A) Probable abandonment windows, $\tau$, are calculated using Eq. 1 for discrete values of $P$ (coloured bars). (B) A continuous probability distribution of $\tau$ is calculated as the density of $P$ within each discrete window of $\tau$ in (A). (C) In reality, $a_{min}$ is not perfectly known, and has an associated age uncertainty that must be accounted for. (i) The ±3σ uncertainty on $a_{min}$ provides a distribution of probable values of $a_{min}$. (ii) The distribution of $a_{min}$ values is discretised. In the Matlab tool, we have set this discretization to be 1/10 the 1σ uncertainty on the youngest age, $a_{min}$, to provide a highly resolved result (note that the cartoon illustration here shows much wider discretisation bins for ease of visualisation), but this discretisation value can be modified. The discrete window of $\tau$ used to calculate the density of $P$ in (B) is set to the same width. (iii) Probability distributions of $\tau$ are calculated for each potential value of $a_{min}$ (as per panel B), and weighted according to the probability distribution of $a_{min}$ values. (iv) The weighted, temporally shifted $\tau$ distributions are then summed to produce a final probability distribution of surface abandonment timing that incorporates uncertainty in the youngest age. (D) Equations used to infer the timing of surface abandonment, calibrated with our artificial data.**





**Figure 8: Two alluvial-fan surfaces in the Laguna Salada Basin, northern Baja California, Mexico. Left: Q4 surface; right: Q7 surface, after Spelz et al. (2008). (A) Locations of surface boulders sampled for [10]Be cosmogenic nuclide exposure dating. (B) Boulder exposure ages recalculated after Spelz et al. (2008) (white circles) and mean surface ages $\pm 1\sigma$ (yellow stars). (C) Probability distributions of $\tau$ calculated using Eq. 1 and incorporating uncertainty on $a_{min}$ following Fig. 7 (black). For illustrative purposes, probability distributions of $\tau$ are shown if uncertainty on $a_{min}$ is not incorporated (red dashed). (D) Selected palaeoclimate proxies: the GRIP ice core $\delta^{18}O$ record from Greenland (blue; Johnsen et al., 1997) and the LR04 global benthic $\delta^{18}O$ stack (black, Lisiecki and Raymo, 2005). Marine Isotope Stages (MIS) are indicated by boxes.**





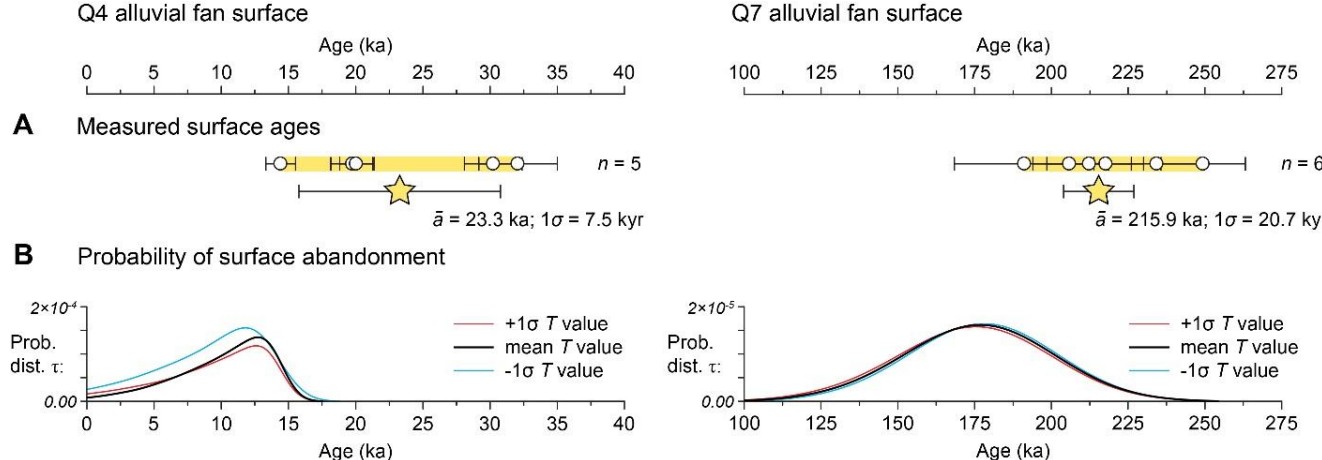

**Figure 9:** (A) Measured surface ages for the Laguna Salada alluvial fan surfaces, following Fig. 8b. (B) Probability distributions of $\tau$ calculated using Eq. 1 and incorporating age uncertainty, where $T$ is derived from the measured spread of surface ages using the mean case in Fig. 6 (black curves) and $\pm 1\sigma$ uncertainties on $T$ (red and blue curves).