# Peer review of "Inferring the timing of abandonment of aggraded alluvial surfaces dated with cosmogenic nuclides"

_Earth Surface Dynamics, 2019_

## Referee Comment (RC1) · Luca C Malatesta (Referee) · 12 Jun 2019

review of D'Arcy et al. 2019 by Luca Malatesta

Dear Editor,

I have read the latest a manuscript by D'Arcy and colleagues with pleasure. They offer a new probabilistic approach to identify the likeliest age of abandonment of an alluvial surface based on series of exposure-dated samples at its surface. They build a power law that predicts the likeliest amount of time elapsed between the youngest surface age and the effective fluvial incision based on the distribution of surface ages

assuming their uniform distribution during the period of activity of said surface. It is a useful contribution that can be applied widely and is definitely worthy of publication in ESurf! In my opinion, the manuscript is ready for publication pending minor clarifying modifications. The article is very well written and easy to read. I would however encourage the authors to consider modifying their probabilistic approach and follow an explicit derivation of their probability power law without requiring the use of "artificial data" for empirical fitting.

Below I briefly describe an alternative approach for the probability law and I provide line by line comments on the text.

Probability The approach using synthetic data has the advantage of mimicking a field situation with n dated boulders out of a larger number. However, it seems to me that using an explicit approach would be much more advantageous. There is no need to graphically fit the powerlaw and deal with the associated error margins, the term "artificial data" can be avoided altogether, and the theoretical framework would be reinforced. Further it would become a more flexible platform, for example to introduce non-uniform distribution of surface ages. I have asked Quentin Berger, probabilist at Paris-Sorbonne, for some help as to how the explicit derivation can be made. I include a document that summarises his explanation hereby. The derivation would replace section 4.1 and provide a definitive and clean solution for this approach. I think it would improve the impact of the manuscript. That being said, it is not a necessary modification and the manuscript stands on its legs as is. It is for the authors to decide whether they want to follow an explicit approach or not.

Line by line

p. 2 L. 31. "These approaches risk circular or inaccurate interpretations." Can you elaborate or give a few examples of these risks?

p. 3 L. 13-15. I suggest to indicate that these ages are arbitrarily selected to produce the scenarios. The reader (or at least I) might think that they are lucky draws from

random rounds and that you are already talking about experiment results. It's a small detail but it would help focusing on the examples you are building.

p. 4 l. 4. "In this study, we use artificial data [...]" At this point it can be unclear whether you use artificial data on virtual surfaces or if you populate a real "geomorphic surface" with artificial data. I suggest to maybe include the purpose of the approach here already: e.g. "we use artificial data to simulate the characteristics of surveyed surfaces" (which you bring up only later at the end of the paragraph on l. 9-10.) This entire paragraph is actually paramount as it frames the use of "artificial data" for the first time. I suggest to carefully edit it such that the combination/coexistence of artificial data and field sites is clear. At this point in the text, Many readers will be asking themselves "ok i understand the problem and motivation but how is that useful for my field site?".

p.4 l. 27. Missing coma after "T"

p. 4 l. 27-28. I suggest to indicate the uniform distribution of the ages here already. The readers might be wondering about it.

p. 5 l. 13. tau = a_min - t_aban is an important relation, I'd suggest to give it a full equation line.

p. 7. The lines of equations lack punctuation.

p. 7 l. 6. "then tau = 12 kyr for P = 0.95." I'd suggest to paraphrase the end of the sentence in plain english for clarity.

p. 7 l. 11-15. the parameter k has a negative value. It should be mentioned here (important for what happens when n grows to infinite). Potentially even better — and I believe in accordance with the convention for such parameters — give k a positive value with an explicit negative sign in the equation.

p. 8 l. 21: section 4.3 is very good and will be very useful.

p. 8. l. 24: using the parameters values listed above I assume? It might be worth

specifying it.

p. 9 l. 11-17. this paragraph reads a little like conclusion material. I am not sure it is necessary.

p. 9 l. 23. "significantly" probably needs defining since you provide a quantity of "one order of magnitude" thereafter.

p. 9, l. 25. There is no figure 5d.

p. 12 l. 10 Without much context, I don't see why that would be a "conundrum".

p. 13 l. 2-4. I am not sure that this characterisation is fair to previous work, many authors showed the importance of timing abandonment and not mean ages. The stand-out "finding" of the present manuscript is to propose a simple and efficient method to get there using incomplete datasets. It's an important step.

Best wishes, Luca Malatesta

Please also note the supplement to this comment:
https://www.earth-surf-dynam-discuss.net/esurf-2019-21/esurf-2019-21-RC1-supplement.pdf

**Supplement:**

Let $X_1, ..., X_n$ be independent variables following the same law $X$, e.g. uniform on $[0, T]$. This example is for a surface where $t_{aban} = 0$.

We look for the minimum value of $X$ (which gives $\tau = X_{min} - 0$) given by the law of the minimum

$$M_n = \min\{X1, ...X_n\}. \tag{1}$$

We then calculate the function

$$F_{M_n}(\tau) = 1 - P(M_n > \tau), \tag{2}$$

where

$$P(M_n > t) = P(x_i > t \text{ for any } 1 \le i \le n) = P(x > \tau)^n, \tag{3}$$

if and only if all $X_i$ are $> \tau$ and independent.

For the uniform law on $[0, T]$:

$$P(X > \tau) = 1 - \frac{\tau}{T}. \tag{4}$$

And so, provided $\frac{\tau}{T} \ll 1$, we get the following exponential repartition function

$$P(M_n > \tau) = \left(1 - \frac{\tau}{T}\right)^n \simeq e^{-\frac{\tau}{T}n}. \tag{5}$$

---

## Referee Comment (RC2) · Anonymous Referee #2 · 21 Jun 2019

In this manuscript, D'Arcy et al. propose a new probabilistic approach to constrain the timing of alluvial surface abandonment using cosmogenic radionuclide dating of surface boulders. Using randomly sampled surfaces ages from a hypothetical alluvial surface, a distribution of surfaces ages are obtained where both the number of samples and age of the depositional surface are varied. The discrepancy between the ages sampled and the true timing of surface abandonment are then determined. The relationships drawn from this analysis are then also applied to an independently dated alluvial fan system in Mexico to infer the timing of surface abandonment. The manuscript is motivated by better constraining the timing of surface abandonment; the authors suggest that this may be a more useful constraint than an average surface age

which is unlikely to relate to any particular forcing or event of interest. In contrast, the timing of abandonment will likely reflect changes in climate, base level change, tectonic forcing or major drainage reorganization. I enjoyed reading this manuscript – it addresses a well thought-out set of questions, is very well written and I believe is a valuable contribution to the field. I would recommend the manuscript for publication pending a few very minor clarifications.

I have a general query about boulders and age distributions and their representation. In the conceptual model, it is assumed that boulders are evenly distributed across the surface and that there is a uniform probability distribution of selectable ages. This is mentioned in the experimental assumptions too (section 3.3). I am curious as to how much these two assumptions are likely to modify the modelling results, and whether these assumptions are actually more likely to be the norm in reality. Is this by any chance something that has been examined or tested? The fact that many alluvial surfaces are not characterized by large numbers of large boulders does indeed suggest that their delivery downstream of their source areas may be temporally clustered and correspond to very large events – this may not be relevant given that it is only the youngest ages which matter here. In general, the authors do a thorough job of highlighting the assumptions and limitations of their approaches.

Section 3.1 – The first time I read this section I was a little confused – it felt like the second paragraph was more observations made from the data rather than a description of methods (line 20-25 in particular). Perhaps some re-phrasing or reordering of material may help with the flow of this section.

Fig. 1B – Could you add a y-axis on the kernel density plot?

P4. L4-11. Again, I had to read this paragraph a couple of times over to work out what was a 'true' timing and a 'real' surface – some confusion on what you have modelled and what is a 'real' example. You also do not mention/introduce that you apply the modelling to a case study in either section 1 or 2. Instead, it does feel like it pops

slightly out of the blue during the paper discussion. Perhaps integrate this into the end of section 1 where you outline what you are going to present with respect to the artificial data and generation of probabilistic equations.

P9. L13 – I don't think it is unreasonable to say that an average age does not/should not correlate with an external forcing.

P.9 L 21. This is probably more for my own curiosity. For the variables you have modelled, you state that 6 to 7 ages are sufficient to characterize the timing of abandonment when T = 30 kyr. You also touch on this in section 5.3 but was wondering if you could just clarify/expand. In your artificial case, the period of surface activity is defined. What if you turn up at a new field site without any indication of how old/period of time each surface has been active for? How many samples are needed/adequate to estimate the timing of surface abandonment to a high degree of probability? Perhaps some idea of the periodicity of forcing mechanism needs to be known (if climatically driven) – but then the argument becomes somewhat circular! Or should we just grab as many samples as we can and state the uncertainty/probability?

P9. L25 – There is no Fig. 5D.

P11. L9 – Should this be Figure 8D?

P12. L12 – If displacement can only occur after surface abandonment, do you have any constraint on a minimum age of displacement onset? Could this estimated time-averaged slip still only be a minimum rate? If so, perhaps state somewhere.

P12. L 28-30 This is a really good point – I also feel that this shouldn't just be in a limitations section! Deriving an average surface age would certainly be biased by burial of older material but by focusing on the timing of abandonment this bias is removed. Perhaps bring this up earlier in the manuscript as an additional strength of this method.

---

## Author Comment (AC1) · 5 Jul 2019

Universität Potsdam • Institut für Geowissenschaften
Karl-Liebknecht-Str. 24-25 • 14476 Potsdam-Golm GERMANY

[Figure]

**University of Potsdam**
**Institute of Geosciences**

Dr Mitch D'Arcy
Alexander von Humboldt Postdoctoral Fellow
mdarcy@uni-potsdam.de

*Telephone:* +49 331 2882 8832
*Date:* 5th July 2019

**Response to reviewer comments of ESurf-2019-21:** "Inferring the timing of abandonment of aggraded alluvial surfaces dated with cosmogenic nuclides"

Dear Dr Malatesta,

Thank you for your detailed and constructive review of our manuscript. We are pleased that you like the essence of our work, and we have responded to each of your comments in detail below. We have incorporated many of your suggested changes and we feel the manuscript is now much improved.

Yours sincerely,

Dr Mitch D'Arcy
(on behalf of all authors)

**Reviewer 1: Dr Luca Malatesta**

Dear Editor, I have read the latest a manuscript by D'Arcy and colleagues with pleasure. They offer a new probabilistic approach to identify the likeliest age of abandonment of an alluvial surface based on series of exposure-dated samples at its surface. They build a power law that predicts the likeliest amount of time elapsed between the youngest surface age and the effective fluvial incision based on the distribution of surface ages assuming their uniform distribution during the period of activity of said surface. It is a useful contribution that can be applied widely and is definitely worthy of publication in ESurf! In my opinion, the manuscript is ready for publication pending minor clarifying modifications. The article is very well written and easy to read. I would however encourage the authors to consider modifying their probabilistic approach and follow an explicit derivation of their probability power law without requiring the use of "artificial data" for empirical fitting.

We thank Dr Malatesta for his helpful review. We are pleased that he likes our work, and we respond to his comments below.

Below I briefly describe an alternative approach for the probability law and I provide line by line comments on the text.

Probability The approach using synthetic data has the advantage of mimicking a field situation with n dated boulders out of a larger number. However, it seems to me that using an explicit approach would be much more advantageous. There is no need to graphically fit the powerlaw and deal with the associated error margins, the term "artificial data" can be avoided altogether, and the theoretical framework would be reinforced. Further it would become a more flexible platform, for example to introduce non-uniform distribution of surface ages. I have asked Quentin Berger, probabilist at Paris-Sorbonne, for some help as to how the explicit derivation can be made. I include a document that summarises his explanation hereby. The derivation would replace section 4.1 and provide a definitive and clean solution for this approach. I think it would improve the impact of the manuscript. That being said, it is not a necessary modification and the manuscript stands on its legs as is. It is for the authors to decide whether they want to follow an explicit approach or not.

We agree that it's worthwhile to consider an analytical solution, and we're grateful that Dr Malatesta and Dr Berger have taken the time to derive these suggested equations. Nonetheless, there are several reasons why these equations cannot replace the approach we take using artificial data.

First, let us quickly summarize how we understand the derivation in the document. Essentially, the probability is evaluated that a particular sample is older than the time of abandonment (set to zero in the document) plus a specified duration $\tau$ (eq. 4). This period $\tau$ is identified with the time difference between the youngest sample and the time of surface abandonment (which is also called $\tau$ in our manuscript). Because we assume uniform distribution of sampled boulders with equal likelihood of sampling, the probability is given by 1- $\tau/T$, where T is the length of surface activity. Next, it is required that all samples fulfil this criterion, and consequently, the probability is raised to the power of n, where n is total number of samples (eq. 5). Finally, assuming $\tau<<T$, the equation is expanded to first order, using a Taylor series, noting that the result is equal to a Taylor expansion of an exponential function to first order.

There are a number of points that can be made in response.

Firstly, the suggested equations are incomplete. A comparison of eq. 5 in the derivation and eq. 2-3 in our manuscript shows that our equations yield more detail, for example a prefactor to the exponential term (Eq. 2) and the dependence of the exponent on the specified percentile (Eq. 3). Further details of

our results, for example the relationship between $T$ and the spread of sample ages (Eq. 6) are not dealt with. Therefore, the derivation may provide a first step, but further steps still need to be worked out.

Secondly, we do not think that the derivation actually reproduces what we are simulating with the artificial-data approach. The youngest sample age in our approach is not older than $\tau$, but determines $\tau$ (i.e., we require a sample with age $\tau$). Thus, the determination of the probability is not correct (Eq. 4 in the derivation). We have since developed some ideas of how to correct the equations, but this is far from giving a usable or publishable result. Whether the exponential approximation that arises from the derivation is coincidental or whether there is a relationship to our approach is not yet clear to us.

Thirdly, even if a complete analytical solution is possible (which it might not be), it seems likely that a numerical solution or artificial data are necessary to provide other elements of a workable approach. An example is the estimation of $T$, where only the span $a_{max} - a_{min}$ can ever be measured empirically (i.e., Fig. 6 and section 4.2). There are actually several advantages to choosing an artificial-data approach, which we mention at the end of section 2. One additional reason that is not mentioned in the manuscript is that the artificial data may be easier to understand for researchers who do not have a rigorous mathematical background. Our equations are correct, the error margins of our parameterisation are very small (see Fig. 5), and importantly, our equations do not require approximations such as $\tau/T \ll 1$ such as in the suggested derivation, which Fig. 4 shows is not realistic.

We are very open-minded about the possibility of developing a full analytical solution in the future, but this is a complicated problem to solve and the suggested equations only provide a starting point. For these reasons, we believe it is advantageous to continue with our approach using artificial data. We decided to not include analytical derivations in the present paper.

Line by line

p. 2 L. 31. "These approaches risk circular or inaccurate interpretations." Can you elaborate or give a few examples of these risks?

Yes, we have clarified the text. We changed "*approaches*" to "*assumptions*", because this sentence is referring directly to the previous sentence where we open this point with two specific examples, including citations. We added an additional citation to Macklin et al. (2002). We now explain in the text that these examples (1) assume that abandonment coincides with palaeoclimate events, and then conclude that climate controls aggradation/incision cycles (risking circularity); and (2) assume that the youngest sampled age approximates abandonment, which our analyses show will often not be the case (risking inaccuracy).

p. 3 L. 13-15. I suggest to indicate that these ages are arbitrarily selected to produce the scenarios. The reader (or at least I) might think that they are lucky draws from random rounds and that you are already talking about experiment results. It's a small detail but it would help focusing on the examples you are building.

Done, this is a good suggestion.

p. 4 l. 4. "In this study, we use artificial data [...]" At this point it can be unclear whether you use artificial data on virtual surfaces or if you populate a real "geomorphic surface" with artificial data. I suggest to maybe include the purpose of the approach here already: e.g. "we use artificial data to simulate the characteristics of surveyed surfaces" (which you bring up only later at the end of the paragraph on l. 9-10.) This entire paragraph is actually paramount as it frames the use of "artificial data" for the first time. I suggest to carefully edit it such that the combination/coexistence of artificial data and field sites is clear.

At this point in the text, Many readers will be asking themselves "ok i understand the problem and motivation but how is that useful for my field site?".

These are good points. We have edited the paragraph to make our approach and its utility clearer, and elaborate on how our synthetic data approach can inform real field studies.

p.4 l. 27. Missing coma after "T"

Corrected.

p. 4 l. 27-28. I suggest to indicate the uniform distribution of the ages here already. The readers might be wondering about it.

We agree that it's important to point out the uniform distribution of selectable ages, but we think a better place to discuss the assumptions of our approach is section 3.3 'Experimental assumptions' (now at p.6, l.12), once the reader is familiar with our overall approach involving artificial data. We now address this particular assumption explicitly in sections 3.2, 3.3, 5.1, and 5.3, which we think is in good context.

p. 5 l. 13. tau = a_min - t_aban is an important relation, I'd suggest to give it a full equation line.

Good idea, we have done this. We put the equation at the start of section 3.1 and re-numbered the other equations accordingly.

p. 7. The lines of equations lack punctuation.

We're not sure what punctuation is missing from the equations. If our article is accepted then we are of course happy for it to be formatted following ESurf style guidelines.

p. 7 l. 6. "then tau = 12 kyr for P = 0.95." I'd suggest to paraphrase the end of the sentence in plain english for clarity.

Done.

p. 7 l. 11-15. the parameter k has a negative value. It should be mentioned here (important for what happens when n grows to infinite). Potentially even better  T and ˘ I believe in accordance with the convention for such parameters  T give k a positive ˘ value with an explicit negative sign in the equation.

We now explicitly point out that *k* has a negative value (p.7, l.34).

p. 8 l. 21: section 4.3 is very good and will be very useful.

Thanks!

p. 8. l. 24: using the parameters values listed above I assume? It might be worth specifying it.

Yes, the parameters (and equations) we derive from our artificial data are universal. Rather than specifying this here, we have added a line, "*Equations 2 through 6 are thus calibrated using our artificial data…*) to the end of section 4.2 above (p. 9, l. 11-11), to make this clear.

p. 9 l. 11-17. this paragraph reads a little like conclusion material. I am not sure it is necessary.

We disagree. This is the opening paragraph of the Discussion and we think it should briefly outline the key implications of our work, namely that abandonment ages will often be more informative than average surface ages, and that our probabilistic approach provides a new way of constraining abandonment. Given that readers thinking about their own field sites will likely jump to this subsection (5.1, "Implications for surface dating"), we think that a very brief discussion of the key points is helpful.

p. 9 l. 23. "significantly" probably needs defining since you provide a quantity of "one order of magnitude" thereafter.

We changed "*significantly*" to "*substantially*"*,* as this sentence is only supposed to be a qualitative statement.

p. 9, l. 25. There is no figure 5d.

This was a typo, we have corrected it to Fig. 5b. Thanks for spotting it!

p. 12 l. 10 Without much context, I don't see why that would be a "conundrum".

We changed "*This conundrum could be partly resolved…*" to "*More realistic values can be obtained…*" (p.13, l.19).

p. 13 l. 2-4. I am not sure that this characterisation is fair to previous work, many authors showed the importance of timing abandonment and not mean ages. The standout "finding" of the present manuscript is to propose a simple and efficient method to get there using incomplete datasets. It's an important step.

We think this is referring to p. 14, l. 2-4, rather than p. 13 (which does not refer to previous work). The majority of studies that date surfaces such as alluvial fans do simplistically represent the surface with an average age (whether a mean, mode, or the peak of a frequency distribution) and rarely attempt to infer the subsequent age of abandonment (although we do explicitly acknowledge several examples in the Introduction). We're certainly not claiming to be the first to consider abandonment, but we do feel it is fair to conclude in our paper that "*the timing of surface abandonment may provide more informative and more precise interpretations than taking an average of measured surface ages*"*,* because one of the novel contributions of our work is quantify the precision with which abandonment can be inferred.

For example, Fig. 4 demonstrates that for desirable probabilities, the timing of abandonment can indeed be pinned down more precisely than the period of surface formation, *T*. Similarly, the example application to the younger Q4 surface on the Baja California fans (Fig. 8, left) shows that surface abandonment likely overlaps with the Younger Dryas, offering a more precise and informative interpretation than the average surface age (for reasons we elaborate on in section 5.2.1). Both of these results demonstrate the value of inferring abandonment, and the potential precision with which this can be accomplished, in a new way that hasn't been demonstrated before.

---

## Author Comment (AC2) · 5 Jul 2019

Universität Potsdam • Institut für Geowissenschaften
Karl-Liebknecht-Str. 24-25 • 14476 Potsdam-Golm GERMANY

[Figure]

**University of Potsdam**
**Institute of Geosciences**

Dr Mitch D'Arcy
Alexander von Humboldt Postdoctoral Fellow
mdarcy@uni-potsdam.de

*Telephone:*    +49 331 2882 8832
*Date:*           5th July 2019

**Response to reviewer comments of ESurf-2019-21:** "Inferring the timing of abandonment of aggraded alluvial surfaces dated with cosmogenic nuclides"

Dear Reviewer 2,

Thank you for your detailed and constructive review of our manuscript. We are pleased that you like the essence of our work, and we have responded to each of your comments in detail below. We have incorporated many of your suggested changes and we feel the manuscript is now much improved.

Yours sincerely,

Dr Mitch D'Arcy
(on behalf of all authors)

**Reviewer 2: Anonymous**

In this manuscript, D'Arcy et al. propose a new probabilistic approach to constrain the timing of alluvial surface abandonment using cosmogenic radionuclide dating of surface boulders. Using randomly sampled surfaces ages from a hypothetical alluvial surface, a distribution of surfaces ages are obtained where both the number of samples and age of the depositional surface are varied. The discrepancy between the ages sampled and the true timing of surface abandonment are then determined. The relationships drawn from this analysis are then also applied to an independently dated alluvial fan system in Mexico to infer the timing of surface abandonment. The manuscript is motivated by better constraining the timing of surface abandonment; the authors suggest that this may be a more useful constraint than an average surface age which is unlikely to relate to any particular forcing or event of interest. In contrast, the timing of abandonment will likely reflect changes in climate, base level change, tectonic forcing or major drainage reorganization. I enjoyed reading this manuscript – it addresses a well thought-out set of questions, is very well written and I believe is a valuable contribution to the field. I would recommend the manuscript for publication pending a few very minor clarifications.

We thank Reviewer 2 for his/her thoughtful review, and we respond to his/her comments below. Reviewer 2 has raised some very interesting questions that we hope will inspire future studies!

I have a general query about boulders and age distributions and their representation. In the conceptual model, it is assumed that boulders are evenly distributed across the surface and that there is a uniform probability distribution of selectable ages. This is mentioned in the experimental assumptions too (section 3.3). I am curious as to how much these two assumptions are likely to modify the modelling results, and whether these assumptions are actually more likely to be the norm in reality. Is this by any chance something that has been examined or tested? The fact that many alluvial surfaces are not characterized by large numbers of large boulders does indeed suggest that their delivery downstream of their source areas may be temporally clustered and correspond to very large events – this may not be relevant given that it is only the youngest ages which matter here. In general, the authors do a thorough job of highlighting the assumptions and limitations of their approaches.

We're pleased that Reviewer 2 thinks we have thoroughly highlighted the assumptions and limitations of our approach, because we want to be upfront about these.

We have not yet performed explicit tests with different distribution shapes of selectable surface ages. One of the reasons for this is simply because it would multiply the analyses in our current manuscript by $x$ number of different distribution shapes, and each would require a substantial amount of consideration (to work through the predictive relationships and equations), and too many figures and analyses for one paper. However, we do agree that it would make a very interesting question for a future study, which could also bring in a compilation of age clusters from well-sampled alluvial fans in order to empirically look at what shapes these distributions tend to have in the real world.

For our work here, the main implications of changing the shape of selectable age distributions would be to (i) change the number of samples needed to get an accurate estimation of $T$; and (ii) change the value of $\tau$ for a given probability/number of ages/$T$. We can speculate here with two cases:

1. **Selectable ages are normally-distributed with a peak in the middle of $T$.**

   A slightly greater number of ages would be needed to estimate $T$ accurately, because you're more likely to end up sampling ages that cluster around the middle of the depositional

timespan, as opposed to distributed randomly throughout $T$ like in our scenarios. Therefore, everything in Fig. 6 would probably be shifted down to slightly lower ratios of $(a_{max} - a_{min})/T$.

Next, $a_{min}$ would presumably fall further from $t_{aban}$ in many cases, which would make $\tau$ slightly larger for a given value of $P$. The magnitude of this effect might in turn depend on $n$ and $T$ (i.e., a bit like Fig. 2). So for small values of $n$ changing the distribution shape might have a bigger effect, but as $n$ increases, perhaps the results would become more similar to ours.

2. **Selectable ages are biased towards the youngest end of $T$ with a long tail decaying towards the older end of $T$.**

   Again, slightly more ages would be needed to estimate $T$ accurately, simply because the sampled ages will always be biased by clustering (wherever the cluster sits within $T$). However in this scenario, you're more likely to densely sample near to the timing of abandonment, which should result in a smaller value of $\tau$ for a given value of $n$, $T$, and $P$, i.e., more accurate estimates of abandonment timing.

So, we speculate that different distributions of selectable ages would result in different effects, probably changing the size of $\tau$ by some small amount. A dedicated study would be needed to pin these effects down. We think that a good way to go would be to start with a compilation of measured ages from natural fans, to see if ages appear to be randomly distributed throughout a timespan, or with a particular distribution shape of the tails.

Section 3.1 – The first time I read this section I was a little confused – it felt like the second paragraph was more observations made from the data rather than a description of methods (line 20-25 in particular). Perhaps some re-phrasing or reordering of material may help with the flow of this section.

We agree that the text needed some clarification here, which was also raised by Reviewer 1. We have edited both section 3.1 and the preceding section 2 (see response to Reviewer 1). We chose to keep the reference to the example case in Fig 1 (referred to as p.4, l.20-25 above) because it illustrates the key advantage of using artificial data and bridges the Justification and the Methods section. However we have added a sentence after this point that explicitly points out why an artificial data approach makes sense, and edited the section to make the text clearer.

Fig. 1B – Could you add a y-axis on the kernel density plot?

In principle we could, yes, but it would be somewhat meaningless because the values would only reflect the size of the x-axis bins used to make the frequency distributions. This is only a cartoon illustration so we feel that would complicate the figure unnecessarily, and we left out the y-axis. Of course, later in the paper the y-axis becomes meaningful when we develop the probabilistic approach, so we do then add y-axis labels.

P4. L4-11. Again, I had to read this paragraph a couple of times over to work out what was a 'true' timing and a 'real' surface – some confusion on what you have modelled and what is a 'real' example. You also do not mention/introduce that you apply the modelling to a case study in either section 1 or 2. Instead, it does feel like it pops slightly out of the blue during the paper discussion. Perhaps integrate this into the end of section 1 where you outline what you are going to present with respect to the artificial data and generation of probabilistic equations.

These are good points. We have edited the paragraph (now p.4, from l.8), also in response to Reviewer 1, to make it clearer. We have rephrased "*real surfaces*" as "*natural surfaces*" for clarity. We also agree about flagging up the case study earlier on; as suggested, we now mention this at the end of section 1.

P9. L13 – I don't think it is unreasonable to say that an average age does not/should not correlate with an external forcing.

This is a good point, this sentence can be phrased in a better way. We have changed (now at p.10, l.16):

> "*…our findings indicate that averages of sampled surface ages are likely to be imprecise representations of the mid-point of surface formation, **and** may not **correlate** with **any** external forcing event…*"

to:

> "*…our findings indicate that averages of sampled surface ages are likely to be imprecise representations of the mid-point of surface formation, **which** may not **coincide** with **a particular** external forcing event…*"

We agree that our results do not explicitly demonstrate that average surface ages will not correlate with external events. They do demonstrate that average ages will often be imprecise representations of the actual average surface age (i.e., Fig. 3a), so we have kept the first part of the sentence. For the second part, we now simply point out that the average surface age might not coincide with external forcing events, for the reasons we discuss in section 2 (Justification). The mid-point of surface formation is, by definition, in the middle of a period of stability when a surface continues being deposited, unlike when the switch from activity to abandonment occurs.

P.9 L 21. This is probably more for my own curiosity. For the variables you have modelled, you state that 6 to 7 ages are sufficient to characterize the timing of abandonment when $T = 30$ kyr. You also touch on this in section 5.3 but was wondering if you could just clarify/expand. In your artificial case, the period of surface activity is defined. What if you turn up at a new field site without any indication of how old/period of time each surface has been active for? How many samples are needed/adequate to estimate the timing of surface abandonment to a high degree of probability? Perhaps some idea of the periodicity of forcing mechanism needs to be known (if climatically driven) – but then the argument becomes somewhat circular! Or should we just grab as many samples as we can and state the uncertainty/probability?

These are great questions, and we have been thinking about this issue of how many samples to collect too! It's true that Fig. 3 is referring to the case where $T = 30$ kyr, but Fig. 4 on the other hand is looking at a wide range of values of $T$, and we still see that the curves really start to level off after about 6-7 samples. The value of $\tau$ is larger when $T$ is larger, but collecting a few extra samples (say, 10) rarely counteracts the effect of increasing $T$. In other words, you don't know what $T$ is when you're sampling in the field, but whatever number of samples you collect it's unlikely to make much of a difference anyway after ~6 or 7 ages (unless you collect hundreds, which is not feasible). Even if $T$ happens to be very large, collecting 10 ages rather than 7 still probably won't be enough to offset the effect of $T$.

We think it's a really interesting idea that climatic periodicity might be driving the formation of fan surfaces, and therefore might provide a guide for the size of $T$. Reviewer 2 is right to point out the risk of circularity, which is why we don't speculate about these questions in this paper, but we are certainly planning to explore this topic in future papers containing data from real alluvial fan systems. We hope our work here inspires other groups to date more fans, because to tackle this question we really need more field examples with well-dated fan surfaces.

Regarding how many samples to collect, our view (based mostly on Fig. 4 and Fig. 6) is that going up to 6 ages will always provide benefits whatever the age/duration of the surface. Collecting more than 6 ages will give smaller and smaller returns, so while it might be useful when especially precise constraints on abandonment are required (e.g., comparing with millennial-scale climate events), it would probably be better to spend those additional resources dating a different surface. The case study from the Baja California fans illustrates this well – the Q4 surface is only dated with 5 ages, but that's still enough to show fairly convincingly that abandonment overlaps with the Younger Dryas (which only lasted ~1 kyr, so is as short as most palaeoclimate events come). Collecting another 5 ages from that surface would narrow the probability distribution a little bit, but it would probably be a better use of time and money to use those 5 samples to date something else.

One caveat here is that an old outlier was discarded from the Q4 dataset (attributed to nuclide inheritance). So if the goal in the field is to measure ~6 'good' ages, then gathering 1 or 2 extra samples might still be useful in case there are some outliers that need to be thrown out. Given unlimited resources, our strategy would be to process 6 or 7 samples per fan surface, but collect another 1 or 2 samples to keep in reserve. Even if 1 or 2 of the ages turned out to be outliers then the dataset would still probably be fine for inferring abandonment timing (for most purposes). If 3+ outliers turned up, or the project required very precise estimates of abandonment age, then you could go back and process the backups.

This is a bit tangential, but if you're going out in the field to sample fans, it's worth taking some free Landsat-8 imagery with you to help choose your sampling sites. We published a paper in *Remote Sensing of Environment* (2018) titled "Alluvial fan surface ages recorded by Landsat-8 imagery in Owens Valley, California", where we talk about these opportunities. Landsat-8 imagery is a really powerful resource when sampling fans and can make a big difference to ensuring you collect samples from the right patches of the surfaces, and ultimately get robust datasets.

P9. L25 – There is no Fig. 5D.

This was a typo, we have corrected it to Fig. 5b. Thanks for spotting it!

P11. L9 – Should this be Figure 8D?

Corrected.

P12. L12 – If displacement can only occur after surface abandonment, do you have any constraint on a minimum age of displacement onset? Could this estimated time-averaged slip still only be a minimum rate? If so, perhaps state somewhere.

We assume that a faulted surface would start to accumulate a displacement as soon as it's abandoned (i.e., as soon as it stops being actively resurfaced). That perspective in turn assumes that a fault is continuously slipping, or at least that the time interval between slip events is insignificantly small compared to the age of surface abandonment. It might be that in some cases there is an additional lag time, which would make the time-averaged slip rate a minimum estimate as Reviewer 2 suggests, even when calculated using the abandonment age instead of the average surface age. However we really don't know whether there is likely to be a lag time in a significant number of cases, so we prefer not to make a general suggestion that time-averaged rates will be minimum estimates. Applied studies will probably need to evaluate this possibility on a case-by-case basis.

P12. L 28-30 This is a really good point – I also feel that this shouldn't just be in a limitations section! Deriving an average surface age would certainly be biased by burial of older material but by focusing on

the timing of abandonment this bias is removed. Perhaps bring this up earlier in the manuscript as an additional strength of this method.

Thank you! We prefer to avoid repetition in different parts of the manuscript, but we agree that this is a valuable point so we have emphasised it more clearly in the text at p.14, l.4-12.

**Additional Edits**

We decided to add a list of mathematical notation (now section 7). This doesn't add much length, but we think it will make it easier for readers to get to grips with our equations.

We have made some very small text edits throughout to improve the wording and clarity of a few sentences. These edits are all shown with tracked changes.

We renamed sub-section 4.3 from '*Application to __real__ surface ages*' to '*Application to __measured__ surface ages*', related to the comments by Reviewer 2 above. We also added a small amount of clarification to this section, just to make sure the text is very clear about how readers could go about applying our approach to their own data (which is ultimately our goal!). For example, here we explain that "*discrete values of $\tau$ can be converted into a probability distribution by calculating the density of P within fixed increments of $\tau$*". We decided that it would be helpful to briefly expand on this point with two additional sentences that clarify what we mean and also explain how the Matlab script can be used to implement our approach. The section is still concise, but we think it will now be easier for readers to follow Fig. 7 and reproduce our approach with their own data.

Related to Fig. 7, we decided to take out the equations because it isn't necessary to reproduce them in the figure and was a waste of space. Figure 7 is now a single-column figure.

We added a citation to Terrizzano et al. (2017) at p.12, l.17.

---

## Author Response (AR2)

Universität Potsdam • Institut für Geowissenschaften
Karl-Liebknecht-Str. 24-25 • 14476 Potsdam-Golm GERMANY

[Figure]

[Figure]

Editors – Earth Surface Dynamics

**University of Potsdam**
**Institute of Geosciences**

Dr Mitch D'Arcy
Alexander von Humboldt Postdoctoral Fellow
mdarcy@uni-potsdam.de

*Telephone:*   +49 331 2882 8832
*Date:*        22nd July 2019

**Revision: esurf-2019-21** "Inferring the timing of abandonment of aggraded alluvial surfaces dated with cosmogenic nuclides"

Dear Prof. Mudd,

Thank you very much for provisionally accepting our article, and for your good advice on improving the supplementary information.

We have now added a second file including 'ReadMe's and guidance for both the Matlab code and also navigating the Excel spreadsheet (as we think both would be helpful for readers). The figures in our article can be reproduced using either the code or the spreadsheet, and we now explicitly explain how this can be achieved in the supplementary PDF. We also added some additional comments in the code, in line with your suggestions. These are helpful additions and we agree they will make it easier for others to reproduce and work with our findings.

We will be delighted to see this work published in ESurf!

Yours sincerely,

Dr Mitch D'Arcy
(on behalf of all authors)